# Characterization and Optimization of Elastomeric Electrodes for Dielectric Elastomer Artificial Muscles

**DOI:** 10.3390/ma13235542

**Published:** 2020-12-04

**Authors:** Guangqiang Ma, Xiaojun Wu, Lijin Chen, Xin Tong, Weiwei Zhao

**Affiliations:** 1School of Mechanical and Electrical Engineering, Xi’an University of Architecture and Technology, Xi’an 710055, China; gq.ma@xauat.edu.cn (G.M.); tongxin@xauat.edu.cn (X.T.); 2School of Mechanical and Electronic Engineering, Wuhan University of Technology, Wuhan 430070, China; lj.chen@whut.edu.cn

**Keywords:** dielectric elastomer actuator, elastomeric electrode, soft robotics, smart materials

## Abstract

Dielectric elastomer actuators (DEAs) are an emerging type of soft actuation technology. As a fundamental unit of a DEA, the characteristics of compliant electrodes play a crucial role in the actuation performances of DEAs. Generally, the compliant electrodes can be categorized into uncured and cured types, of which the cured one commonly involves mixing conductive particles into an elastomeric matrix before curing, thus demonstrating a better long-term performance. Along with the increasing proportion of conductive particles, the electrical conductivity increases at the cost of a stiffer electrode and lower elongation at break ratio. For different DEA applications, it can be more desirable to minimize the electrode stiffness or to maximize its conductivity. In examination of the papers published in recent years, few works have characterized the effects of elastomeric electrodes on the outputs of DEAs, or of their optimizations under different application scenarios. In this work, we propose an experimental framework to characterize the performances of elastomeric electrodes with different formulas based on the two key parameters of stiffness and conductivity. An optimizing method is developed and verified by two different application cases (e.g., quasi-static and dynamic). The findings and the methods developed in this work can offer potential approaches for developing high-performance DEAs.

## 1. Introduction

Dielectric elastomer actuators (DEAs), an emerging type of soft actuation technology, demonstrate advantages over conventional actuators in terms of large actuation strain, high energy density, fast responses, and low cost [1,2]. A fundamental DEA consists of a piece of dielectric elastomer (DE) sandwiched between two compliant electrodes, as illustrated in Figure 1. When actuated, the DEA expands in area and contracts in thickness. The most widely adopted DE materials include acrylic (e.g., 3M VHB 4910 [3,4] and silicone (e.g., Wacker ELASTOSIL 2030 [5,6], Wacker RT 625 [7]).

Despite the fact that acrylic DE has a high quasi-static actuation strain due to the low elastic modulus and high dielectric constant, its high viscoelasticity leads to a slow response speed and a reduced dynamic output. In contrast, silicone elastomers show a significantly lower viscoelasticity, hence an improved dynamic output performance with a peak output power density (~600 W/kg) higher than natural muscles reported to date [8]. The promising applications of silicone-based DEAs have been widely demonstrated in the literature, including a soft pump [9,10], soft crawling robot [11], flapping wing micro-air-vehicles [12] and a versatile soft gripper [13].

In addition to the DE material that affect the actuator’s performance, compliant electrodes (CE) also play a critical role in determining the DEA’s outputs. An ideal CE needs to have high conductivity and low elastic modulus so as not to impede the movement of DEA [14]. Furthermore, good adhesion to the DE film is necessary to ensure that the CE actuates together with the DE and there is no delamination in practical applications. In general, the compliant electrodes can be categorized into uncured and cured types, with the uncured types being mainly composed of carbon grease and loss powder. For example, Matysek et al. adopted the technique of directly spraying carbon black (CB) onto the DE film [15]. However, the CB particles can detach from the DE film during actuation, especially during high-frequency vibrations, hence the lifespan of the DEA is limited. Carbon grease can be easily applied to the DE film using brushes or scratch [16] and can offer good adhesion with the film [17,18]. However, the oil can volatile slowly over time, which limits the lifespan of the DEA. Furthermore, this kind of uncured electrode is likely to delaminate when touching, thereby resulting in contamination and damage of the electrode pattern. Thus, its practical application is limited. The cured electrodes are commonly elastomeric electrodes which are based on CB particles due to their good conductivity and low cost. Compared with carbon nanotubes, CB is cheaper and the fabrication process is simpler as the ultrasonic pre-dispersion is not required [19]. Typical elastomeric electrodes are a mixture of CB and uncured silicone elastomers. Once applied to the DE film, the electrodes can fully cure and adhere to the DE film, which overcomes the limitations of the uncured electrodes. The amount of conducting particles is the main focus of the reported papers [20,21]. Rosset et al. [22] assessed the degradation of elastomeric electrodes and reported a low lifespan of only a few thousand cycles. Meanwhile, Saint-Aubin et al. [23] studied the life of the carbon-based electrodes, and found that aging (as seen in the change in resistance and in actuation strain versus cycle number) is independent of the actuation frequency. Schlatter et al. [24] developed a CB-based elastomeric electrode which shows low resistance and stiffness and the influence of the elastomeric electrode thickness was investigated. Zhang et al. investigated the impacts of the ambient temperature [25], humidity [26], thickness and conductivity of electrode [27] on the DE performance.

Despite the development of the elastomeric electrodes reported in the literature, systematic investigation on the effects of elastomeric electrodes on the output performance of the DEAs is seldom found. In addition, to the best of the authors’ knowledge, no work has developed optimization methods for the elastomeric electrode formulas. The performance of an elastomeric electrode can be controlled by the two principal factors of inherent stiffness and conductivity, which affect the DEA’s responsive strain and electrical response speed, respectively. In this work, we characterize the stiffness and conductivity of varying elastomeric electrode formulas. Based on the characterization results, we then develop a method for optimizing the electrode formula. Examples of optimal electrode formulas are given for different application scenarios including quasi-static actuation and dynamic resonant actuation. The findings reported and the methods developed in this work can offer potential approaches for developing high-performance DEAs.

## 2. Materials and Methods

### 2.1. Materials

ELASTOSIL 2030 silicone film (Wacker, Munich, Germany), which is widely commercially available, was adopted as the DE material. The main silicone rubber for elastomeric electrode fabrication is RT 625 (Wacker, Munich, Germany), which has low viscoelasticity and high elongation at break ratios and has thus been widely adopted for DEA applications [28,29]. However, RT 625 shows a high stiffness (tensile strength of 6.50 N/mm^2^ compared to 6.00 N/mm^2^ for ELASTOSIL 2030), to accommodate this, soft Ecoflex 20 (Smooth-on, Macungie, PA, USA) with the tensile strength of 1.10 N/mm^2^ was added to the matrix. Carbon black powder with the diameters of 35–45 nm was adopted as the conductive material (XFI15, XFNAN, Shuzhou, China). Isopropanol (AR, Dong Jiang, Shanghai, China) was added to dilute electrode compounds during the mixing stage, which ensures an even dispersion of the CB in the compound. Sil-poxy (Smooth-on, Macungie, PA, USA) was used to adhere the silicone film to the metal membrane holder. The fabricated DEA film and its support frame were bonded with tape (5302A, NITTON, Kita-ku, Japan).

### 2.2. Preparation of Specimen

In this tensile experiment, the shape of the sample used is in accordance with the international standard ASTM D412 protocol [30]. We choose the ‘Die C’ sample shape with an overall length of 115 mm in which the narrow section is 33 mm long. The sample template was fabricated by cutting polyethylene glycol terephthalate (PET) with a thickness of 0.1 mm by a laser cutting machine (VLS2.30, Universal, DC, USA). The PET template was placed on top of an acrylic plate (Hongwang, Shenzhen, China) then the well-mixed CB-silicone mixture was added to the PET template and then scratched with a scraper. The CB mass ratios, α, was varied from 10 wt.% to 25 wt.% with a step of 5 wt.% and the Ecoflex 20-to-RT 625 ratios of β = 0, 0.2:1, 0.4:1, 0.6:1, 0.8:1 were tested, which gives a total of 20 combinations. For each combination, three samples were fabricated and tested to eliminate random errors. The detailed fabrication process is described for the sample with α = 10 wt.% and β = 0.2:1 and schematic illustrations of the fabrication process are given in Figure 2a. A total of 0.5 g CB was added to 5 g of isopropanol and then mixed at 3000 rpm for 10 min using a mixer (DACC150R, Speed-Mixer, Shanghai, China). After mixing, 3.75 g RT625 part AB (1A:9B) and 0.75 g Ecoflex20 (1A:1B) were added and the mixture was again mixed at 3000 rpm for another 8 min. The resulting mixture was scraped onto the template PET plate then placed in a 60 °C constant temperature oven (XGQ-200, JLG, Guangdong, China) for 12 h to cure completely.

The DEA fabrication process is shown in Figure 2b. Firstly, the silicone elastomer was stretched biaxially by 1.1 × 1.1 then attached to a ring shape membrane holder. Compliant electrodes were then patterned on the DE film by pad printing. The thickness of the electrode with the machine (DEKTAK-XT, BRUKER, Karlsruhe, Germany) was about 13 μm. Finally, the DEA film was bonded to an acrylic frame with adhesive tape to maintain its tension.

### 2.3. Testing Methods

#### 2.3.1. Electrical Conductivity Test

The electrode samples for electrical conductivity measurement were fabricated using the same method described above. The sample configuration is a rectangular strip with a length of 100 mm and a width of 10 mm (as is shown in Figure 3a). 

The high aspect ratio L/w = 10:1 adopted in this work ensures a homogeneous current flow, as suggested in the reference [31]. A four-point probe method was adopted here with a multimeter (E4980AL-102, Keysight, CA, USA) following the protocols from the literature [31]. The strip was in contact with a metal electrode covering the whole width of the strip at each end (as illustrated in Figure 3b). For an electrode of thickness d, length L and width w, made of a homogenous material of volume resistivity ρ, the volume (or bulk) electrical resistance, R, along the length is given by [31]
(1)$$R=\rho \frac{L}{w\;d}$$

The surface (or sheet) electrical resistance $\;R_{s}$ of the electrode is defined as:(2)$$R_{s}=R\frac{w}{L}=\frac{\rho }{d}$$

The surface resistance of the portion of the sample delimited by the length L can thus be calculated as follows:(3)$$R_{s}=R\frac{w}{L}=\frac{V}{I}\frac{w}{L}$$
where V and I are the voltage and current across the two ends of the electrodes, respectively. 

The electrical conductivity test was also performed to assess the homogeneity of the electrodes. A sample with the CB ratio of α = 20 wt.% and pure RT 625 was prepared, and its volume resistance from for values of L/w varying from 5 to 10 was measured.

#### 2.3.2. Uniaxial Tensile Test and Dielectric Elastomer Actuator (DEA) Actuation Test

The tensile tests were carried out using a tensile testing machine (ESM 303, MARK-10, Copiague, NY, USA), which was connected to a computer via the universal serial bus (USB) for data collection. The fabricated electrode specimen was attached on the testing machine at both ends, where a linear motor stretched one end of the specimen at a constant velocity of 49.9 mm/min. A load cell measured the reaction force of the stretched electrode. The tensile test setup is illustrated in Figure 4a and photographs of the setup are shown in Appendix A in the Appendix A. The ambient temperature was kept constant at 24 °C and the relative humidity was measured at 40%.

In order to determine the reliability of the tensile test protocol, three samples with the same electrode formula combination were tested under the same testing conditions. The expression of the statistical dispersion can be calculated with a type A evaluation of measurement uncertainty (i.e., standard deviation of the sample mean $s_{\;\bar{x}}$):(4)$$s_{\;\bar{s}}=\frac{s}{\sqrt{n}}$$

With s being the corrected sample standard deviation:(5)$$s=\sqrt{\frac{1}{n}\overset{n}{\underset{i=1}{\sum }}{\left(y_{i}-{\;\bar{y}}_{i}\right)}^{2}}$$
where n is the number of samples, $y_{i}$ is the value of the true stress measured, and ${\;\bar{y}}_{i}$ is the arithmetic mean of the repeated true stress measured.

In the next test, three samples with different ratios were fabricated and the mean curve and samples’ standard deviation obtained from three tensile tests of the three samples are plotted. The measurement results found that the materials based on this fabrication method have an advantage in consistency and repeatability. Appendix A illustrates the tensile test experimental results for three specimens with the Ecoflex 20: RT 625 mixture ratio of 0.2:1. The identical curves demonstrate a good consistency and repeatability in both the fabrication and tensile tests. 

To investigate the influences of the electrode conductivity and stiffness on the DEA’s output performances, quasi-static and dynamic tests were conducted on DEA samples with different electrode formulas. A conical DEA geometry was configured in these tests. To test the quasi-static performance, a 0.5 N mass was hung directly on the center of the DEA to form a conical configuration, as shown in Figure 4b. A sinusoidal wave with the frequency of 0.2 Hz and an amplitude of 4132 *V* was applied to the DEA by a high voltage amplifier (10/40A-HS-H, TREK, Lockport, NY, USA) and a laser displacement sensor (LK-G153, Keyence, Osaka, Japan) measured the actuation stroke of the DEA.

The same platform was used on high frequency testing (Figure 4c). Using a linear spring deforms the center of the DEA film out-of-plane by 4 mm. Frequencies sweeping from 0 to 150 Hz at a rate of 1 Hz/s were generated by MATLAB (MathWorks, Natick, MA, USA) and amplified using the high voltage amplifier prior to being applied to the DEA. The laser displacement sensor recorded the real-time position of the DEA at a sampling frequency of 2 kHz.

## 3. Results

### 3.1. Morphology Characterization

The microscope (Ci-Pol, NIKON, Tokyo, Japan) was used for the specimen morphological characterization. These samples (the batch that was used in the conductivity measurement) were pure RT625 with different ratios of carbon black (i.e., 5%, 10%, 15%, 20%, 25%, 30%). Figure 5 shows the distribution of CB ratio in the substrate respectively. The images with CB ratio of 25% and 30% were amplification with a higher ratio in the upper right corner. The microscope images show that in the samples with a CB ratio of less than 30% no cracks formed on the elastomeric electrodes. As the CB ratio is increased to 30%, cracks can be observed.

### 3.2. Electrical Characterization

The measured conductivity of the electrodes made with different formulas are plotted in Figure 6. Figure 6a plots the measured volume resistance against the measuring distance. It can be noted that, as the measuring distance increases, the measured volume resistance increases in an approximately linear way, which agrees well with the theory described in Equation (1), for this reason demonstrating a good homogeneity of the electrode fabricated in this work. In Figure 6b, it can be seen clearly that a higher content of CB leads to a better electrical conductivity, particularly when the CB mass ratio *α* is greater than 15%. It is also noteworthy that adjusting the ratio of Ecoflex 20 and RT 625 *β* has a negligible effect on the electrical conductivity to within 5% error.

### 3.3. Mechanical Characterization

Figure 7 plots the measured strain-stress relationship of the elastomeric elastomers made with different formulas, as well as the gradients of each of these strain-stress curves. It shows that the electrodes can be stretched by more than four fold its original length before breaking at the lowest CB ratio of α = 10 wt.%. As the CB ratio α increases, the specimen breaks at a lower stretch length, and as the value of α is increased to 25 wt.%, the material can only be stretched by less than 100 % before it breaks. It should also be noted that as the Ecoflex 20-to-RT 625 ratio β increases, the electrodes become softer. However, there is no clear trend in the effects of β on the elongation at break ratios. Figure 8 characterizes the tensile modulus (defined as the mean slope before a strain-stress of 40%) of each electrode formula combinations. It can be seen the modulus increases with the CB ratio α but in the meantime decreases with the increasing β due the lower tensile strength of Ecoflex 20 compared to RT 625.

## 4. Discussion

The conductivity of the elastomeric electrode was measured in Section 3. Figure 6 shows that the conductivity of the elastomeric electrode depends on the mass fraction of conductive CB. As the mass fraction of conductive carbon black increases, the conductivity of the flexible electrode increases. However, the pursuit of electrical conductivity leads to poor ductility of the elastomeric electrode. At the same time, the hardness also increases non-linearly. These trends can be gathered from Figure 7.

While the addition of Ecoflex 20 can reduce the stiffness of the elastomeric electrodes, it can also reduce the bonding strength between the silicone elastomer and the electrodes. A qualitative bonding test was performed to demonstrate the adhesion between the elastomeric electrodes and the DE film through choosing the two combination α = 20 wt.%, β = 0.2:1 and α = 20 wt.%, β = 1:1 (The bonding test results for other ratios are shown in Appendix A in the Appendix A). It can be seen from Figure 9a that the elastomeric electrode of α = 20 wt.%, β = 0.2:1 is tightly attached to the DE film part when pulled by a silicone tape. When the tape is removed, no obvious electrode residue can be seen on the tape and the electrode remained intact without any signs of delamination. However, the elastomeric electrodes of α = 20 wt.%, β = 1:1 shows clear delamination when tested with the same methods (shown in Figure 9b). No delamination was observed for the electrodes with the Ecoflex 20-to-RT 625 ratio of β = 0, 0.2:1, 0.4:1, 0.6:1, 0.8:1. As a result, an Ecoflex 20-to-RT 625 ratio greater than 0.8:1 is not recommended for this formula. Meanwhile, the cracks indicate that (1) the CB particles have affected the formation of long polymer chains, and (2) high CB ratios were no longer suitable for elastomeric electrodes. To sum up, we chose the following parameter values of α = 10 wt.%, 15 wt.%, 20 wt.%, 25 wt.%, β = 0, 0.2:1, 0.4:1, 0.6:1, 0.8:1.

The stiffness and conductivity of the electrodes were analyzed separately on the output performances of the DEAs. Tables are provided to show their formulations, the conductivity (Table 1) and the modulus (Table 2). The α = 10 wt.% and α = 20 wt.% were selected according to 10-fold the conductivity. Because the α = 25 wt.% breaks at a lower stretch length. The formula of α = 20 wt.%, β = 0.2:1 and α = 20 wt.%, β =0.6:1 were selected because of the similar conductivity. Meanwhile, the formula α = 10 wt.%, β =0.2:1 was selected based on the similar stiffness. To sum up, the testing was by using the following three different combinations: Formula (1): α = 10 wt.%, β = 0.2:1; Formula (2): α = 20 wt.%, β = 0.2:1 and Formula (3): α = 20 wt.%, β =0.6:1. The conductivity and modulus of three different combinations are shown in Table 3.

Figure 10a shows that, in quasi-static application scenarios (in this case, frequency is 0.2 Hz, and the output voltage was 4132 V as shown in Figure 10c, the actuation stroke of Formula 1 is the highest, followed by Formula (3), and then Formula (2). It is worth noting that, despite the conductivity being the lowest, the lowest tensile modulus of Formula (1) still leads to the highest actuation stroke, which proves that the stiffness of the electrodes has a greater impact on the DEA’s performance in quasi-static cases. However, as is shown in Figure 10b, in high-frequency dynamic applications, the peak output stroke of the DEA with Formula (3) is greater than Formulas (2) and (1). This is due to the high electrical conductivity of the Formula (3) electrodes which leads to a much faster electrical response speed, hence the DEA can be charged and discharged fully at high frequencies. It should also be noted that the resonant frequency of the DEA using Formula (2) is higher than that using Formulas (1) and (3), because of its high stiffness. Nonetheless, conductivity is the dominant factor for determining the actuator’s performances in high-frequency application scenarios. 

The electrode formula optimization method is provided based on the above characterization. For quasi-static application scenarios, the stiffness influence is significantly larger than the conductivity (recall that the conductivity for Formula (1) is over 10-fold lower than the other two, yet it demonstrates the highest stroke in quasi-static cases). Based on this, we recommend a type of electrode with a lower stiffness: α = 10 wt.%, β = 0.8:1. However, for dynamic application scenarios, conductivity has a greater impact on DE film, so the conductivity weight is higher. In this case, the formula of α = 25 wt.%, β = 0.8:1 is suggested.

These findings can be critical for developing high-performance DEA-based robots. For example, in quasi-static actuation scenarios where large actuation strain is required, DEAs can offer promising potentials in bio-inspired swimming robots [32,33] and soft infrared-reflecting systems [34]. For dynamic applications where high resonant amplitudes are demanded, DEAs can be adopted in soft pumps [9,35] and flapping wing robots [8].

Adding Ecoflex 20 to the electrode formulas can not only reduce the stiffness of the electrodes, it was also observed to significantly increase the pot life of the electrode compound during the fabrication process. As illustrated in Appendix A in the Appendix A at the same humidity and temperature, the mixture of Ecoflex20 and RT 625 remains in consistent liquid form after two hours of curing time. However, the electrode solution with pure RT 625 solidifies with a consistency of paste with chunks of solids, as shown in Appendix A in the Appendix A. The long pot life formula of Ecoflex 20 and RT 625 developed in this work would be beneficial for screen printing, spray coating, spin coating, pad printing and casting processes since it can improve manufacturing efficiency and reduce costs.

## 5. Conclusions

Compliant electrodes are a critical component for DEAs as they enable the charge flows of the DEA and deform together with the DE film. Among the most widely adopted CB-based electrodes, elastomeric electrodes can overcome the delamination problem of the other types of carbon electrodes and are thus promising for practical applications in soft robotics and wearable devices. However, to the best of the authors’ knowledges, no works have investigated the influence of elastomeric electrode stiffness and conductivity on the output performance of DEAs in different application scenarios. The performance of an elastomeric electrode can be described by the two main factors of inherent stiffness which affect the DEA’s actuation strain, and its conductivity which determines the actuator’s electrical response speed. This paper is the first to systematically characterize and optimize the two properties of elastomeric electrode materials by analyzing the effects of CB ratio and different silicone mixture ratios. The key findings of this work can be summarized as follows:Increasing the CB ratio increases the electrical conductivity of the electrode and the stiffness of the elastomeric electrode but reduces the elongation at break ratio.Increasing the Ecoflex 20 to RT 625 ratio can decrease the stiffness of the elastomeric electrode yet shows no clear effects on the elongation at break ratio.Adding Ecoflex 20 to the electrode formula can significantly increase the pot life of the electrode mixture.In quasi-static actuation scenarios, electrode stiffness plays the most important role in DEA’s output performance.In high-frequency actuation scenarios, electrode conductivity is the key factor determining the DEA’s output performance.

The research contributions are as follows:The research is the first to analyze the influence of electrode stiffness and conductivity on DEA output performances in different application scenarios.The research developed an optimization method for elastomeric electrode formulas based on different application scenarios (e.g., α = 10 wt.%, β = 0.8:1 for quasi-static and α = 25 wt.%, β = 0.8:1 for high-frequency cases).

The research will promote the development of high-performance DEAs in soft robotics, for example, dynamic locomotion, soft wearable devices and soft energy harvesters. The reported findings and the developed methods in this work can offer potential approaches for developing high-performance DEAs.

## Figures and Tables

**Figure 1 materials-13-05542-f001:**
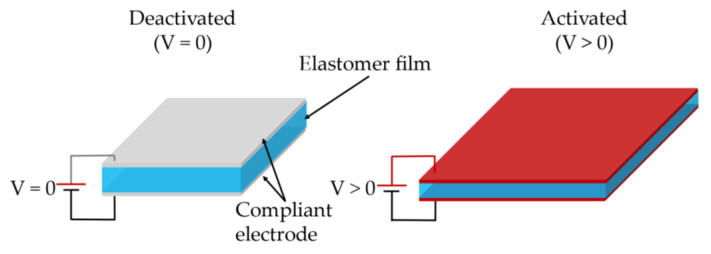
Working schematic of dielectric elastomer actuator (DEA).

**Figure 2 materials-13-05542-f002:**
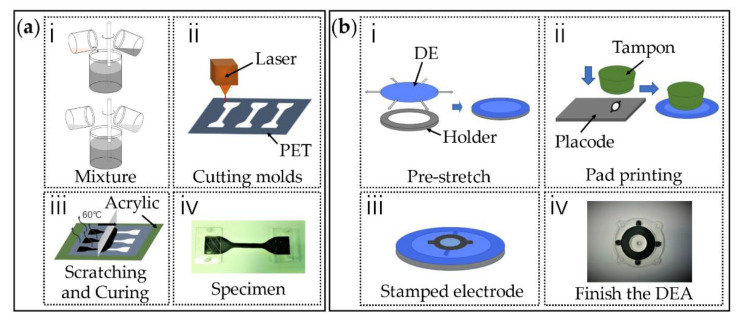
(**a**) Uniaxial tensile test specimen fabrication steps: (**ⅰ**) mixing the carbon black (CB) in isopropanol, then mixing in Ecoflex 20 and RT 625 silicone; (**ii**) laser cutting the PET templates; (**ⅲ**) scratching the compliant electrodes and curing for 12 h; (**ⅳ**) finished specimen. (**b**) DEA specimen fabrication steps: (**ⅰ**) pre-stretching DE film and attaching the DE film to the membrane holder; (**ⅱ**) transferring compliant electrode into the DE film by pad printing; (**ⅲ**) stamping electrode on the DE film; (**iv**) finished DEA.

**Figure 3 materials-13-05542-f003:**
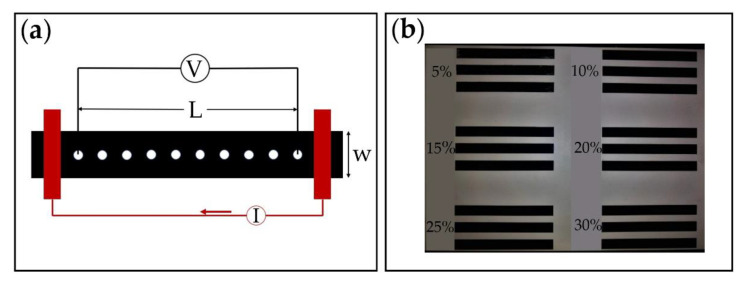
Electrical conductivity test. (**a**) Fabricated specimens; (**b**) schematics of four-point probe test.

**Figure 4 materials-13-05542-f004:**
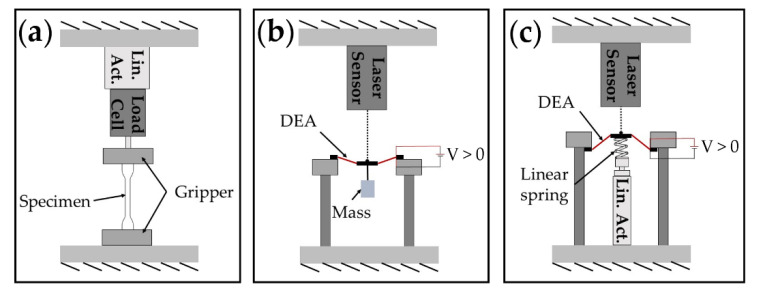
Illustrations of the experimental setups. (**a**) Tensile test; (**b**) quasi-static DEA actuation test; (**c**) high-frequency test.

**Figure 5 materials-13-05542-f005:**
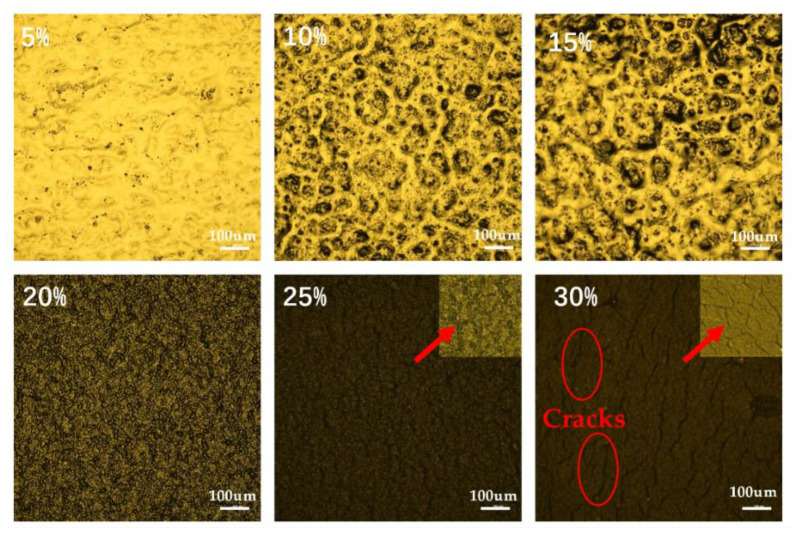
Microscope images of the elastomeric electrodes with different carbon black concentrations. The images with CB ratio of 25% and 30% were amplification with a higher ratio in the upper right corner.

**Figure 6 materials-13-05542-f006:**
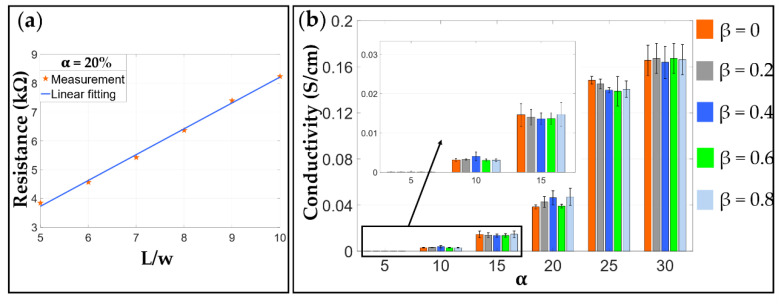
Measured conductivity. (**a**) An approximate linearly increase in the resistance with the measuring distance L/w is observed proving the homogeneity of the fabricated electrodes; (**b**) the conductivity of elastomeric electrodes with different formulas.

**Figure 7 materials-13-05542-f007:**
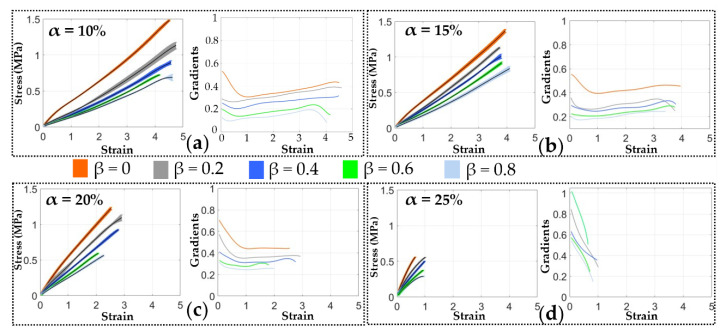
Uniaxial tensile test results of the electrodes with different CB ratios (with 95% confidence bands). (**a**) α = 10 wt.%; (**b**) α = 15 wt.%; (**c**) α = 20 wt.%; (**d**) α = 25 wt.%.

**Figure 8 materials-13-05542-f008:**
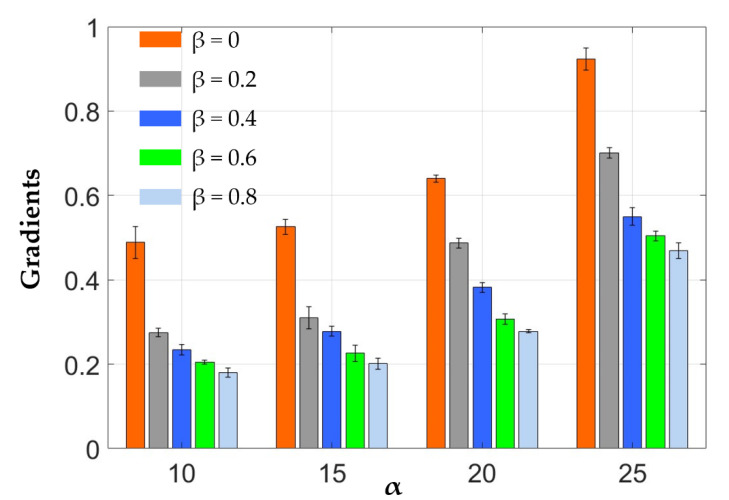
The tensile modulus of the different electrode formula combinations.

**Figure 9 materials-13-05542-f009:**
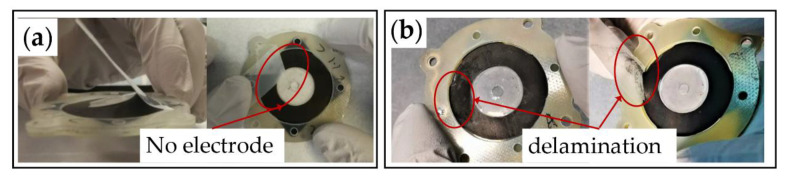
Qualitative bonding experiment: (**a**) α = 20 wt.%, β = 0.2:1; (**b**) α = 20 wt.%, β = 1:1.

**Figure 10 materials-13-05542-f010:**
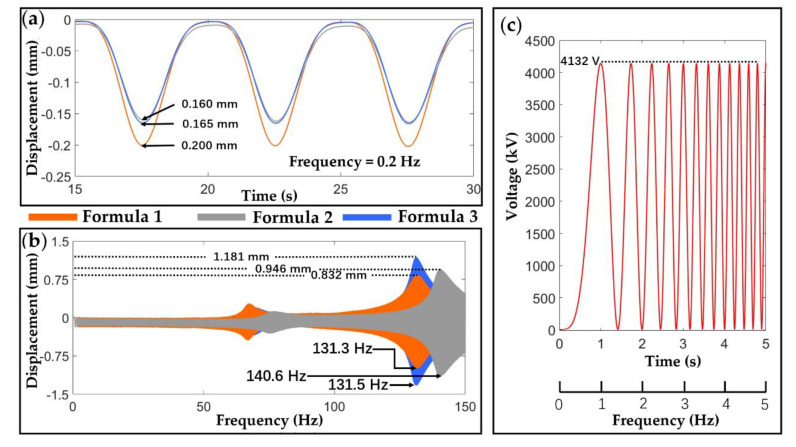
DEA output performance characterization. (**a**) Quasi-static (0.2 Hz) outputs of the DEA samples; (**b**) frequency sweep (0 to 150 Hz) outputs of the same DEAs; (**c**) input voltage.

**Table 1 materials-13-05542-t001:** The conductivity of elastomeric electrodes with different formulas.

Conductivity (S/cm)		α			
		10%	15%	20%	25%
β	0	0.0031 ± 0.0004	0.0145 ± 0.0035	0.0384 ± 0.0021	0.1482 ± 0.0041
	0.2	0.0032 ± 0.0002	0.0139± 0.0024	0.0427 ± 0.0058	0.1449 ± 0.0051
	0.4	0.0039 ± 0.0013	0.0135± 0.0018	0.0463 ± 0.0075	0.1397 ± 0.0027
	0.6	0.0030 ± 0.0004	0.0135± 0.0019	0.0401 ± 0.0019	0.1387 ± 0.0157
	0.8	0.0030 ± 0.0003	0.0146± 0.0037	0.0471 ± 0.0090	0.1404 ± 0.0086

**Table 2 materials-13-05542-t002:** The tensile modulus of the different electrode formula combinations.

Tensile Modulus (MPa)		α			
		10%	15%	20%	25%
β	0	0.489 ± 0.038	0.526 ± 0.018	0.640 ±± 0.008	0.923± 0.026
	0.2	0.285 ± 0.010	0.310 ± 0.026	0.487 ± 0.011	0.701 ± 0.013
	0.4	0.234 ± 0.012	0.278 ± 0.012	0.382 ± 0.012	0.550 ± 0.021
	0.6	0.205 ± 0.005	0.226 ± 0.019	0.297 ± 0.012	0.504 ± 0.011
	0.8	0.180 ± 0.011	0.201 ± 0.013	0.278 ± 0.004	0.469 ± 0.019

**Table 3 materials-13-05542-t003:** The detailed information of three formulas.

Samples	α	β	Conductivity (S/cm)	Tensile Modulus (MPa)
Formula (1)	10	0.2:1	0.0032	0.285
Formula (2)	20	0.2:1	0.0427	0.487
Formula (3)	20	0.6:1	0.0401	0.297

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
