# Peer review of "Characterization and Optimization of Elastomeric Electrodes for Dielectric Elastomer Artificial Muscles"

_materials, 2020, doi:10.3390/ma13235542_

Round 1
Reviewer 1 Report
This paper reports the experimental results regarding mechanical and conductivity properties of carbon black reinforced silicone elastomers. These soft composites were then used as the electrode in dielectric elastomer actuators and their actuation performance was characterized as a function of matrix polymer mix and carbon black ratio. The paper is well written, but the authors must address the following comments before publication:
Line 76: "few works have developed ..." Please provides references and explain the novelty and differences of your work compared to the previous works.
Line 122: Do you have an estimate of the thickness of the electrode formed on the DE film?
Line 164: I don't think ref (28) is relevant to formula (4). Please double check.
Line 189: Which samples have been studied here? The actuator assembly?
Figure 5: Could you explain the difference in morphology of samples as CV percentage increases? Why are the images for 25% and 30% so different? Is it more related to light adjustment in the microscope? Haven't you observed any cracks in 25%?
Figure 6: How do you explain the significance increase in conductivity as CB ratio changes from 10% to 25%, but minor increase when CB ratio increase from 25% to 30%?
Line 259: stiffness of 1 and 3 are the same, so how can it be claimed that stiffness is the dominant factor? Please clarify. Also, has this been reported for other types of DEAs?
Line 263: Again, conductivity of 2 and 3 is the same, so how can we conclude that conductivity is the dominant factor here? Please clarify.
Figure 10: How does these results compare to other types of DEAs? In terms of level of actuation at low and high frequencies.
Reviewer 2 Report
This paper overviews dielectric elastomer actuators (DEAs) performance accounted to combination and variety of stiffness and conductivity values of elastomeric electrodes. The results suggest that the increase of carbon black additive in DE film leads to a better conductivity and stiffness of the material but reduces its elongation at break ratio. Moreover, varying ratios of two different silicon rubber can influence stiffness of the sample. The developed optimization method introduces solutions for quasi-static and high frequency cases of material use. The originality of this work lays in developing optimization method for characterizing DEAs through stiffness (actuation strain) and conductivity (actuator’s electrical response speed) parameters. The objective and hypothesis are stated clearly but the results and discussions are intuitive and lack theoretical backup. Hence, this paper does not provide extensive explanation of observed phenomena.
The article requires additional spelling and punctuation check and partially is written in a confusing language. The sectioning can be recombined more efficiently. The methods used in the study should include detailed explanation. Other researcher should be able to experimentally reproduce the material and repeat the testing.
Specific comments:
· The paragraph under 2.3.3 “Morphology Characterization” seem to have results and discussions included to methods section which assumingly belongs to 3. “Results” and/or 4. “Discussion” section.
· The paragraph that starts with 236 line draws conclusions on absence of electrodes and delamination in the sample is based purely on “naked-eye” observation. Is it enough to draw such conclusions?
· Line 192 statement and microscope images do not match. Some cracks can be seen on 25% sample as well. What is happening to the sample morphology at 10% and 15%?
· The optimization method is based on 20 samples. Is that enough data to predict behavior of other materials and make a suggestion of providing “guidelines for the research of stretchable electrode materials”?
· A representable information on how combinations of alpha and beta ratios for stiffness and conductivity formulas were selected is required and has to be explained. Otherwise, the selection is confusing. Can we really call it optimization method?
· The abstract claims that some particular application is going to be presented in this paper. Even though two cases of DEA output performance were discussed, the application suggestions were very vague. There is an exact application named in the title, but it’s never really being mentioned in the body of the paper.
· Line 64 “isn’t” is suggested to be written as “is not’, a “.” should follow [19] reference, line 125 a “,” has to be included before “then” in the figure 2 caption …
Reviewer 3 Report
This work highlights and discusses the balance between conductivity and stiffness, which is an important issue with regards to compliant electrodes (CE) for dielectric elastomers actuators. They show that increasing the concertation of carbon black (CB) filler in the elastomeric matrix increased its conductivity and its stiffness, which is exactly as would be expected. CEs were fabricated from RT 625 with added Ecoflex 20, which gave some control over the stiffness of the CE. The authors give two example applications areas and suggest the optimal RT 625 : Ecoflex 20 : CB ratios for these examples. Below are questions and recommendations for the authors.
- In equations 1, 2 and 3, the letter R is used to denote different things. Please modify this (e.g. with subscripts) so readers can distinguish between them.
- In Figure 7, it is not clear which sample is which in the ‘Gradients vs Strain’ plots. I believe these should be modified so that the colours match the stress vs strain curves.
- In line 48, ‘critical role’ is repeated. This is one of several grammatical errors throughout the manuscript, please address.
- In the methods section, only Ecoflex to RT625 ratios of 0.8:1 re mentioned, though 1:1 was tested for adhesion to the DE. Can the authors suggest a basis for the increased delamination observed when using higher quantities of Ecoflex? Was the adhesion strength steadily reduced with added Ecoflex, or did it suddenly drop above a threshold ratio of 0.8:1?
- The discussion of the different formulations and their mechanical and electrical properties in lines 250-256 is quite confusing. Can the authors instead include a table, showing the formulations, their modulus, and their conductivity?
- In Figure 10a, the colours do not match the formulas given – I believe the orange trace should be red?
- The most interesting results are given in Figure 10a, with reference to the table suggested in comment 5, can the authors give values for the modulus and conductivity of these formulations, so we can better understand how big an impact this trade-off has in quasi-static scenarios. For this example and the high frequency example, quantitative data on the modulus and conductivity must be given to understand the results and support the conclusions of this manuscript.
- The authors suggest that adding Ecoflex 20 increased the pot life of the electrode compound. Is there a molecular of physical basis for this that he authors can suggest? And is the related to the increased delamination from the DE observed when using a 1:1 ratio of Ecoflex 20 to RT 625?
Round 2
Reviewer 1 Report
I thank the authors for providing response to my comments and revising the manuscript. I do not have any further comments.
Reviewer 3 Report
The manuscript has been significantly improved since its original submission. I belive it is now suitable for publication in Materials.